# The First Complete Chloroplast Genome of *Campanula carpatica*: Genome Characterization and Phylogenetic Diversity

**DOI:** 10.3390/genes14081597

**Published:** 2023-08-07

**Authors:** Won-Sub Yoon, Chang-Kug Kim, Yong-Kab Kim

**Affiliations:** 1Department of Mechanical Design Engineering, Wonkwang University, Iksan 54538, Republic of Korea; 2Genomics Division, National Institute of Agricultural Sciences, Jeonju 54874, Republic of Korea; chang@korea.kr; 3Department of Information Communication Engineering, Wonkwang University, Iksan 54538, Republic of Korea

**Keywords:** *Campanula carpatica*, *Campanula* species, chloroplast genome, phylogeny

## Abstract

*Campanula carpatica* is an ornamental flowering plant belonging to the family Campanulaceae. The complete chloroplast genome of *C. carpatica* was obtained using Illumina HiSeq X and Oxford Nanopore (Nanopore GridION) platforms. The chloroplast genome exhibited a typical circular structure with a total length of 169,341 bp, comprising a large single-copy region of 102,323 bp, a small single-copy region of 7744 bp, and a pair of inverted repeats (IRa/IRb) of 29,637 bp each. Out of a total 120 genes, 76 were protein-coding genes, 36 were transfer RNA genes, and eight were ribosomal RNA genes. The genomic characteristics of *C. carpatica* are similar to those of other *Campanula* species in terms of repetitive sequences, sequence divergence, and contraction/expansion events in the inverted repeat regions. A phylogenetic analysis of 63 shared genes in 16 plant species revealed that *Campanula zangezura* is the closest relative of *C. carpatica*. Phylogenetic analysis indicated that *C. carpatica* was within the *Campanula* clade, and *C. pallida* occupied the outermost position of that clade.

## 1. Introduction

The Carpathian bellflower (*C. carpatica*) belongs to the family Campanulaceae and is one of the more than 300 species in the *Campanula* genus. *C. carpatica* is a herbaceous plant native to Eastern Europe and has bell-shaped purple flowers [1]. This plant is being cultivated worldwide as an ornamental flowering plant due to its big purple flowers [2]. Studies on seed germination and induction of adventitious shoots in vitro in *C. carpatica* have been carried out [3,4]. *Campanula* species are important horticultural crops that show remarkable morphological and habitat diversity [5], and there is an increasing demand for new traits such as flower color, flower shapes, and disease resistance [6].

The plastid genome (plastome) is widely used for the genetic identification and phylogenetic analysis of plant species [7]. The plastome provides variable sites for genetic identification and evolution analyses because it has the advantages of no recombination, the presence of single-copy genes, and a low rate of nucleotide substitution [8]. Complete plastome sequencing is a possible solution for resolving taxonomic relationships in plant species [9]. Plant chloroplasts are the main sites of photosynthesis and play an important role in many cellular functions such as carbon fixation, energy conversion, and stress responses [10]. Chloroplast genomes have specific characteristics such as small size and frequent polymorphisms. They are haploid and exhibit uniparental inheritance [11]. Therefore, chloroplast genomes are widely used in studies of genome evolution, population structure, and molecular genetics [12,13]. The chloroplast genome sizes of higher plants range from 120 to 180 kb [14], and the genome sequences are highly conserved comprising 110–130 genes [15]. The chloroplast genome is highly stable in higher plants, and its circular structure is primarily conserved and divided into four major regions: large single-copy (LSC) and small single-copy (SSC) regions separated by two inverted repeats (IRs) of equal length [16]. However, differences in chloroplast genome sequences can be significant among the species and, therefore, have been useful in species circumscription [17].

In this study, we assembled and annotated the complete chloroplast genome of *C. carpatica* using de novo next-generation sequencing (NGS) and reference-guided assembly. We described the basic characteristics of the genome, such as gene content, genome structure, and repeat sequences, and analyzed phylogenetic relationships among the *Campanula* species.

## 2. Materials and Methods

### 2.1. Plant Material and Chloroplast Genome Sequencing

*C. carpatica* was collected from the National Institute of Agricultural Sciences in Wanju, Korea. Genomic DNA was isolated from fresh leaves, and its integrity, DNA purity, and concentration were analyzed using 1% agarose gel electrophoresis and a NanoDrop spectrophotometer (Thermo Scientific, Waltham, MA, USA). Libraries were prepared by constructing Illumina paired-end (PE) and Oxford Nanopore Technologies (ONT) libraries according to the manufacturer’s instructions. The Illumina PE library was sequenced on an Illumina HiSeq X platform (Illumina, San Diego, CA, USA). The ONT library was prepared on a Nanopore GridION platform (Oxford Nanopore Technologies, Oxford Park, UK).

### 2.2. Chloroplast Genome Assembly and Annotation

Raw reads obtained for *C. carpatica* were mapped to the reference genome of *C. punctata* (NCBI accession number NC_033337). Self-correction was performed on the mapped reads, and de novo assembly was performed using Canu [18]. ONT and Illumina reads were polished using Pilon [19] and Medaka (https://github.com/nanoporetech/medaka/, accessed on 15 September 2022). The Pilon program was used to calibrate the draft genome with Illumina short reads, and the Medaka program was used to calibrate the draft genome with ONT reads. Polishing was performed in various ways, including gap filling, indels of bases in the genome, block substitution, and single-base differences. The chloroplast genome sequence was annotated using the GeSeq tool [20] and manually corrected using the Artemis annotation tool [21] with NCBI BLASTN searches. The architecture of the *C. carpatica* chloroplast genome was visualized using the OGDRAW program [22]. The integrated chloroplast structure map, including cis- and trans-splicing genes, was visualized using the CPGView web tool [23].

### 2.3. Repeat Sequence

Simple sequence repeats (SSRs), long tandem repeats (i.e., size of repeat unit ≥ 7), and dispersed repeats were searched in the *C. carpatica* chloroplast genome. First, SSRs were identified using the MISA-Web software (https://webblast.ipk-gatersleben.de/misa/, accessed on 28 March 2023) with the following parameters: at least ten repeat units for mononucleotides, six repeat units for di-nucleotides, five repeat units for tri-nucleotides, and four repeat units for tetra-, penta-, and hexa-nucleotide repeats. Compound SSRs were defined as repeats that were interrupted by a non-repeat sequence of a maximum of 100 nucleotides. Tandem repeats were detected using the Tandem Repeats Finder software (https://tandem.bu.edu/trf/, accessed on 28 March 2023). Tandem repeats observed in at least one copy were considered significant. The parameters were set as follows: alignment parameters of matches = 2, mismatches = 2, indels = 7, minimum alignment score = 50, maximum period size = 500, and maximum tandem repeat array size = 2. Finally, dispersed repeats were identified using the Vmatch tool with two types (i.e., the direct and the palindromic type) on the 30 bp minimal repeat size (https://github.com/genometools/vstree/; accessed 28 March 2023).

### 2.4. Comparison of Campanula Chloroplast Genomes

Four chloroplast genome sequences were available for the *Campanula* genus in NCBI GenBank (accessed on 20 January 2023): *C. pallida* (NC_063742), *C. punctata* (NC_033337), *C. takesimana* (NC_026203), and *C. zangezura* (NC_057269). Comparative analyses of pseudogene content, percentage identity, and gene content were undertaken to determine the similarities between *C. carpatica* and the four *Campanula* species. The mVISTA program [24] displayed global sequence alignments of genomic sequences from different *Campanula* species. Multiple genome alignments were performed using the Mauve program [25] with default parameters. The IRScope tool [26] evaluated the expansion/contraction of the junction sites within *C. carpatica* and the four reported *Campanula* species.

### 2.5. Phylogenetic Analyses

To infer the phylogeny of *C. carpatica*, we used complete chloroplast genome sequences of 15 Campanulaceae species and *Helianthus annuus* of the Asteraceae family as an outgroup. The following 16 complete chloroplast genome sequences were downloaded from the NCBI GenBank database (accessed on 6 April 2023): *Adenophora triphylla* (MT649408), *C. carpatica* (OP677559), *Campanula pallida* (NC_063742), *Campanula punctata* (NC_033337), *Campanula takesimana* (NC_026203), *Campanula zange-zura* (NC_057269), *Centropogon nigricans* (NC_035761), *Codonopsis lanceolate* (MH251613), *Dialypetalum floribundum* (NC_035357), *Grammatotheca bergiana* (NC_036095), *Hanabusaya asiatica* (NC_024732), *Lobelia muscoides* (NC_035379), *Platycodon grandiflorus* (NC_035624), *Siphocampylus krauseanus* (NC_035760), *Trachelium caeruleum* (NC_010442), and *H. annuus* (NC_007977). These sequences were concatenated using only shared protein-coding sequences among the 16 species. Multiple genome alignments among the 16 species were performed using the MAFFT program [27] to reconstruct the phylogenetic tree. To confirm the phylogenetic position of *C. carpatica* within the Campanulaceae family, we generated a maximum likelihood (ML) phylogenetic tree with 1000 bootstrap replicates using the MEGA11 software [28].

## 3. Results and Discussion

### 3.1. Chloroplast Genome Sequencing and Assembly

A total of 2.4 million mapped reads were assembled into the *C. carpatica* chloroplast genome (NCBI accession number, OP677559). The *C. carpatica* chloroplast genome is 169,341 bp long and has a typical circular structure with an LSC region of 102,323 bp, an SSC region of 7744 bp, and a pair of 29,637 bp inverted repeats (IRa/IRb). The length of *C. carpatica* was within the range reported for higher plant species, that is, 120–180 kb [14]. A total of 120 genes were predicted in the genome, including 76 protein-coding genes, 36 transfer RNA genes, and 8 ribosomal RNA genes. Finally, 100 unique genes were selected from the 120 predicted functional genes after excluding 20 duplicated genes (Table 1 and Figure 1). Twenty duplicated genes were located in the IR region (seven protein-coding genes, four rRNA genes, and nine tRNA genes). Ten cis-spliced genes (*atpF*, *rpl2*, *rpl16*, *petD*, *petB*, *ycf3*, *rpoC1*, *ndhA* (×2), and *ndhB*) and one trans-spliced gene (*rps12*) were identified in the *C. carpatica* chloroplast genome. The *ycf3* gene contained two introns, whereas the remaining nine cis-spliced genes contained only one intron (Appendix A).

Plant chloroplast genes can be functionally categorized into three groups: genes associated with self-replication and protein synthesis, genes involved in photosynthesis, and a third group of “other genes” [29]. The *C. carpatica* chloroplast had 53 self-replication expression-related genes, 43 photosynthetic genes, and 4 other genes (Table 1).

### 3.2. Repeat Sequence Analysis

Three types of repetitive sequences were detected in the *C. carpatica* chloroplast genome: 14 SSRs, 94 long tandem repeats, and 49 dispersed repeats. The chloroplast genome size was affected by differences in the lengths of repetitive sequences. Chloroplast genomes with two IR regions show many rearrangement events in the Campanulaceae species [30]. Repeat sequences lead to gene recombination, genomic rearrangement, genetic diversity, and sequence divergence [31]. Fourteen SSRs were identified in the *C. carpatica* chloroplast genome (Appendix A). A total of fourteen SSRs were composed of ten mononucleotide repeats and four compound repeats, and the mononucleotide A/T repeat unit was most abundant in the chloroplast genome. Four compound SSRs were detected with lengths ranging from 49 to 126 bp. The SSRs, known as microsatellites, are among the most informative molecular markers for studying genetic diversity [32]. Compound SSRs are more polymorphic than single SSRs and are generally regarded as highly mutable loci in the chloroplast genome [33]. In total, 94 long tandem repeats (size of repeat unit ≥ 7) (Appendix A) and 49 dispersed repeats, which are of two types, i.e., 31 direct repeats (forward repeats) and 18 palindromic repeats (Appendix A), were detected in *C. carpatica*. Tandem repeats can serve as genetic markers to unravel population processes in plants [34]. Dispersed repetitive DNA sequences, scattered throughout the chloroplast genome, are the main cause of genome rearrangements and play a major role in genomic sequence variation [35,36]. The 157 detected repeats of these three types can be used as potential molecular markers in the *Campanula* species.

### 3.3. Comparison of Chloroplast Genomes among Campanula Species

Multi-alignment analysis using the mVISTA program was performed to determine the level of divergence between *C. carpatica* and four other *Campanula* species: *C. pallida* (NC_063742), *C. punctata* (NC_033337), *C. takesimana* (NC_026203), and *C. zangezura* (NC_057269). DNA sequence divergence among the species is a function of neutral, deleterious, and advantageous mutation rates [37]. In the sequence divergence of five *Campanula* species, multi-alignment results indicated that the genome structure, gene order, and gene content have highly conserved sequences among the five *Campanula* species. A comparison of the LSC, IR, and SSC regions revealed that the LSC region contained more divergent genes (Figure 2).

Multiple genome alignment is one of the most basic tools used in comparative genomics, and various chloroplast genomes have been used to construct multigene alignments using locally collinear blocks (LCBs) [38,39]. To characterize the structure and collinearity of *C. carpatica* species, we aligned the collinear blocks to generate a whole-genome alignment using the Mauve software. Whole-genome alignment identified four LCBs in five *Campanula* species. The LCBs show major rearrangements because they are connected to the same color-lines in the alignment. The alignment results showed similarities among the four LCBs identified using color-display blocks (Figure 3). Mauve alignment revealed four LCBs, suggesting the presence of two breakpoint genes (*rps12* and *ycf1*) in the IR region. These breakpoints were formed because the chloroplast genomes had positional differences in each genome location. Although the comparison revealed small structural rearrangements in the chloroplast genomes of the five *Campanula* species, they occurred in the IR region (Figure 3).

### 3.4. Contraction and Expansion of Border Regions

The expansion and contraction of the border regions in IRs are important factors for variation in the chloroplast genome [40,41]. The LSC/IR and SSC/IR borders in the chloroplast genomes of the five *Campanula* species were compared (Figure 4). The *ycf2*, *trnL*, *rrn16*, *psbA*, and *trnH* were detected near the LSC/IR border, whereas *psaC*, *ndhG*, and *ndhE* were detected at the SSC/IR border. The IR region sizes of the five *Campanula* species ranged from 26,632 bp to 29,742 bp. The *ndhE* gene was located at the SSC/IRa (JSA) region in all the chloroplast genomes, whereas its corresponding pseudogene *ndhE* fragment (158–164 bp) copy was found at SSC/IRb (JSB) except for *C. pallida*. The *trnH* sequences were observed in the LSC regions of all chloroplast genomes, and this gene was located 117–120 bp away from the IRa/LSC border region. Two copies of the *rrn16* genes were observed in *C. pallida*, *C. punctata*, *C. takesimana*, and *C. zangezura*. The *rrn16* genes were located in the IRa and IRb regions near the LSC/IR borders; however, two copies of *rrn16* genes in *C. carpatica* were not observed in the near LSC/IR and SSC/IR borders. Additionally, the *ycf2* gene was mainly located in the LSC region of other *Campanula* species, whereas *C. carpatica* did not contain the *ycf2* gene.

### 3.5. Phylogenetic Analysis

To resolve the phylogenetic relationships of *C. carpatica*, an ML tree was constructed using 63 shared protein-coding genes from the chloroplast genomes of 15 species of the Campanulaceae family, including one outgroup species from the family Asteraceae.

Chloroplast genomes have been widely used to explore phylogenies because of their low rate of nucleotide evolution, absence of recombination, and uniparental inheritance [42,43]. The topology of the phylogenetic tree was consistent with the traditional morphology-based taxonomy of Campanulaceae species. Our phylogenetic reconstruction was concordant with previously published results [44,45], and the outgroup species, *H. annuus*, had the most distinct characteristics (Figure 5). Most Campanulaceae species clustered with 100% bootstrap values, and *C. carpatica* was the closest relative to *C. zangezura*. Although phylogenetic relationships among the species of the Campanulaceae family are well supported, they do not clearly show the evolutionary relationship because of the low number of chloroplasts in the *Campanula* species. Nevertheless, our phylogenetic analysis is the first to clarify the phylogenetic position of *C. carpatica* within the family Campanulaceae.

## 4. Conclusions

The *C. carpatica* chloroplast genome was sequenced and assembled for the first time. This demonstrates that the combination of Illumina and ONT sequence libraries is sufficient for a high-quality chloroplast genome assembly. Detailed characteristics of the *C. carpatica* chloroplast genome were determined based on analyses of chloroplast features, repeat sequences, sequence divergence, and boundaries between plastome regions. The genome has a typical circular structure, is 169 kb long, and contains 120 functional genes. Genome size, gene content, genomic composition, and phylogenetic relationships were similar to those of other *Campanula* species. Overall, we revealed the complete chloroplast genome sequence of *C. carpatica* and identified 157 repetitive sequences as potential molecular markers for the Campanulaceae family. These findings enrich our knowledge of gene composition, genome evolution, and genetic diversity of the Campanulaceae species. In addition, these genetic resources provide a reference for further genomic studies on the Campanulaceae species.

## Figures and Tables

**Figure 1 genes-14-01597-f001:**
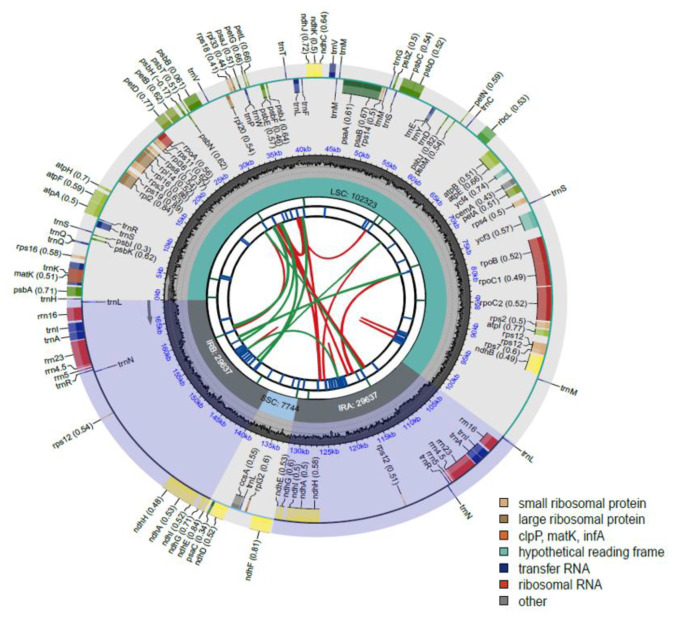
Schematic map of the *C. carpatica* chloroplast genome. The map consists of six circle tracks. From the center outward, the first circle shows the dispersed repeats connected with red (forward direction) and green (reverse direction) arcs. The second circle indicates the tandem repeats, and the third circle indicates the SSR sequences. The fourth circle shows the LSC, SSC, and IR regions. The fifth circle shows the GC content, and the sixth circle shows the genes having different colors based on their functional groups. The inner and outer colored boxes present transcribed clockwise and counterclockwise genes, respectively.

**Figure 2 genes-14-01597-f002:**
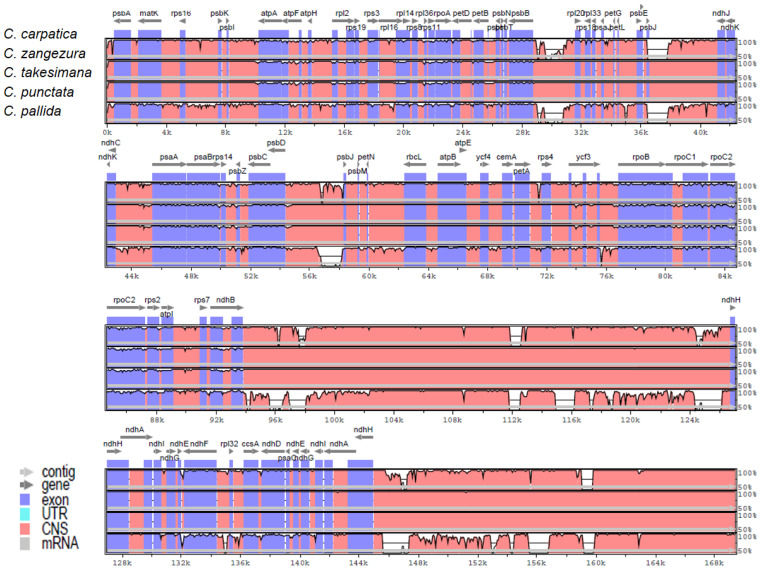
Chloroplast nucleotide sequence alignments between *C. carpatica* and the four *Campanula* species using the mVISTA program. The gray arrows indicate the orientation of genes, and the scale of the *y*-axis represents the percent sequence identity between 50 and 100%. Annotated genes appear at the top, and dissimilar regions are in white.

**Figure 3 genes-14-01597-f003:**
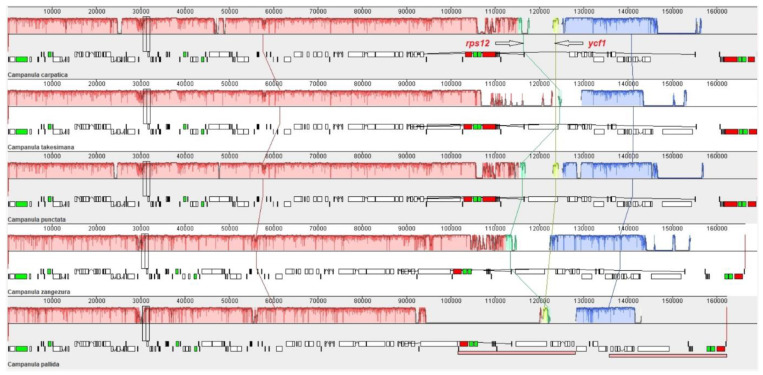
Collinear alignment between *C. carpatica* and the four other *Campanula* species using Mauve software. Locally collinear blocks (LCBs), the most conserved collinear regions of the genomes, are depicted as shaded colored boxes connected by a line, and the *x*-axes indicate the coordinate lengths in the located chloroplasts.

**Figure 4 genes-14-01597-f004:**
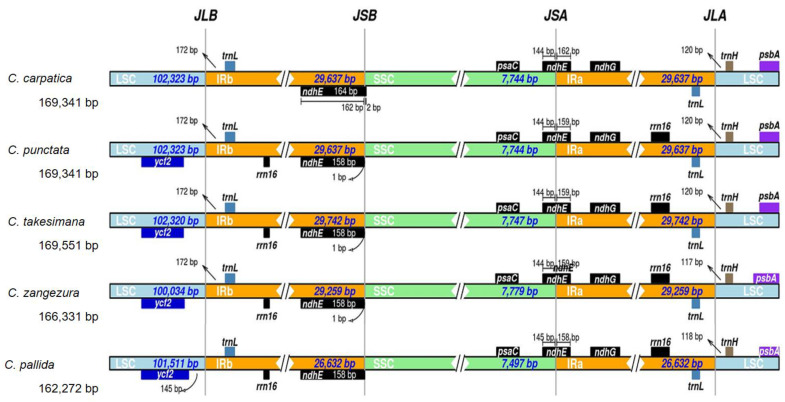
Comparison of boundaries between the LSC, SSC, and IR regions of chloroplast genomes of selected *Campanula* species. JLB, the junction between the LSC/IRb; JSB, the junction between the SSC/IRb; JSA, the junction between the SSC/IRa; and JLA, the junction between the LSC/IRa. Boxes above or below the main lines indicate genes near the described boundaries.

**Figure 5 genes-14-01597-f005:**
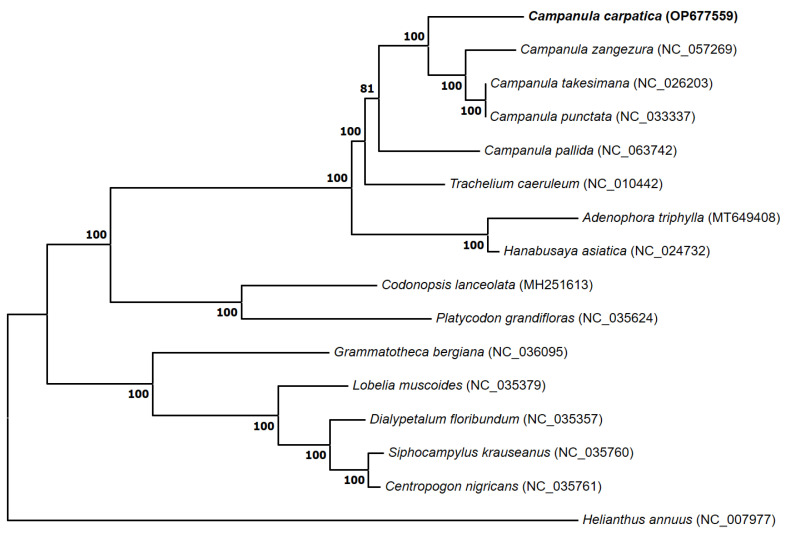
A phylogenetic tree constructed using 63 conserved protein-coding genes of 16 plant species, including the chloroplast genome of *C. carpatica*. The complete chloroplast sequence of the Asteraceae family (*H. annuus*) was used as an outgroup to root the tree. The bootstrap support values (>50%) from 1000 replicates are indicated on the nodes.

**Table 1 genes-14-01597-t001:** Genes in the chloroplast genome of *C. carpatica*.

Category	Group of Genes	Name of Genes
Self-replication	Ribosome (LSU ^1^)	*rpl2*, *rpl14*, *rpl16*, *rpl20*, *rpl32*, *rpl33*, *rpl36*
	Ribosome (SSU ^2^)	*rps2*, *rps3*, *rps4*, *rps7*, *rps8*, *rps11*, *rps12 **, *rps14*, *rps16*, *rps18*, *rps19*
	RNA polymerase	*rpoA*, *rpoB*, *rpoC1*, *rpoC2*
	rRNA genes	*rrn4.5S **, *rrn5S **, *rrn16S3 **, *rrn23S **
	tRNA genes	*trnH-GUG*, *trnK-UUU*, *trnQ-UUG **, *trnS-GCU*, *trnS-CGA*, *trnR-UCU*, *trnV-GAC*, *trnP-UGG*, *trnW-CCA*, *trnT-UGU*, *trnL-UAA*, *trnD-GUC*, *trnC-GCA*, *trnF-GAA*, *trnG-GCC*, *trnS-UGA*, *trnE-UUC*, *trnY-GUA*, *trnS-GGA*, *trnL-CAA **, *trnA-UGC **, *trnR-ACG **, *trnI **, *trnN-GUU **, *trnV-UAC*, *trnL-UAG*, *trnM-CAU ***
Photosynthesis	Photosystem I	*psaA*, *psaB*, *psaC*, *psaJ*
	Photosystem II	*psbA*, *psbB*, *psbC*, *psbD*, *psbE*, *psbF*, *psbH*, *psbI*, *psbJ **, *psbK*, *psbM*, *psbN*, *psbT*, *psbZ*, *ycf3*
	Cytochrome ^3^	*petA*, *petB*, *petD*, *petG*, *petL*, *petN*
	ATP synthase	*atpA*, *atpB*, *atpE*, *atpF*, *atpH*, *atpI*
	NADH ^4^	*ndhA **, *ndhB*, *ndhC*, *ndhD*, *ndhE **, *ndhF*, *ndhG **, *ndhH **, *ndhI **, *ndhJ*, *ndhK*
	Subunit of rubisco	*rbcL*
Other genes	c-type cytochrom ^5^	*ccsA*,
	Envelop membrane protein	*cemA*,
	Maturase	*matK*
	Conserved ^6^	*ycf4*

^1^ Large subunit, ^2^ small subunit, ^3^ cytochrome b/f complex, ^4^ NADH-dehydrogenase, ^5^ c-type cytochrome synthesis gene, ^6^ conserved open reading frames, * genes with two copies, and ** genes with four copies.

## Data Availability

All raw sequencing data produced in this study have been deposited in the NCBI Sequence Read Archive (SRA) under the BioProject (assembly and annotation of the complete organellar genome of *C. carpatica*) number PRJNA957583, BioSample SAMN34257940, and SRA database SRR24223267.

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
