# Peer review of "The First Complete Chloroplast Genome of Campanula carpatica: Genome Characterization and Phylogenetic Diversity"

_genes, 2023, doi:10.3390/genes14081597_

Round 1
Reviewer 1 Report
Due to the lack of line numbering, my comments are incorporated into the attached manuscript file as notes placed close to a certain point of the text.

Moderate editing of the English language is required. Some fragments are hard to follow or sentences are clumsy. In my Review report, I tried to focus on the merit of the paper, not the Language style.
Author Response
Dear reviewer,
We thank the reviewer for the critical and insightful suggestions to improve our study. According to the reviewer’s comments, we have added many sentences and corrected descriptions in the all sections. In addition, revised manuscript checked by Editage (www.editage.co.kr) for English language editing.
Sincerely Yours,
Yong-Kab Kim, Professor
Department of Information Communication Engineering
Wonkwang University
Iksan 54538, Korea
Email: [email protected]

Reviewer 2 Report
I have carefully reviewed the manuscript titled "The first complete chloroplast genome in Campanula carpatica: genome characterization and its phylogenetic diversity" submitted to Genes Journal. Overall, the paper presents an important contribution to the field of genetics and is suitable for publication after addressing the following concerns:
Language: The language of the manuscript is clear and fluent. The authors have effectively communicated their research findings.
Methods and Analyses: The methods and analyses employed in the study have been appropriately planned and described. The authors have utilized next-generation sequencing techniques to elucidate the chloroplast genome of Campanula carpatica and investigate its genetic relationships.
Alignment with Journal's Concept: The manuscript aligns well with the concept of Genes Journal, as it focuses on the genetic characterization and diversity of Campanula carpatica's chloroplast genome.
Introduction and Discussion: However, both the introduction and discussion sections of the manuscript appear to be weaker compared to the other sections. The introduction should provide a general overview of next-generation sequencing techniques and explain why the particular technique used in this study was chosen. Additionally, I recommend expanding the discussion by including more references, and considering the inclusion of comparisons with other ornamental or horticulture plants. It would also be beneficial to provide information on how the findings of this study could be utilized in practical applications.
Conclusion: The conclusion section could be further elaborated to highlight the potential benefits of this study for future research endeavors. Specifically, the authors should discuss how their work can contribute to and guide future studies in the field.
In conclusion, I find this manuscript to be valuable and deserving of publication in Genes Journal, given that the aforementioned concerns are adequately addressed. The improvements suggested above will enhance the clarity, scientific impact, and overall quality of the paper.
Author Response

(The authors gave the same response as above.)

Round 2
Reviewer 1 Report
Dear Authors, I appreciate your efforts in improving your manuscript. However, I found a few elements which need your attention.
Due to the fact that the manuscript lacks line numbering, I decided to use Notes placed close to the certain piece of the text to make it easier to follow

Author Response
Dear reviewer,
We thank the reviewer for the insightful suggestions to improve our study. According to the reviewer’s comments, we have revised in the all sections.
Sincerely Yours,
Yong-Kab Kim, Professor
Department of Information Communication Engineering
Wonkwang University
Iksan 54538, Korea
Email: [email protected]
